# Three-Dimensional-Printed Oral Films Based on LCD: Influence Factors of the Film Printability and Received Qualities

**DOI:** 10.3390/pharmaceutics15030758

**Published:** 2023-02-24

**Authors:** Tingfeng Xu, Huijie Li, Yi Xia, Sheng Ding, Qingliang Yang, Gensheng Yang

**Affiliations:** College of Pharmaceutical Science, Zhejiang University of Technology, Hangzhou 310014, China

**Keywords:** 3D printing, oral films, liquid crystal display, bilayer films, oral mucosal drug delivery

## Abstract

As an oral mucosal drug delivery system, oral films have been of wide concern in recent years because of their advantages such as rapid absorption, being easy to swallow and avoiding the first-pass effect common for mucoadhesive oral films. However, the currently utilized manufacturing approaches including solvent casting have many limitations, such as solvent residue and difficulties in drying, and are not suitable for personalized customization. To solve these problems, the present study utilizes liquid crystal display (LCD), a photopolymerization-based 3D printing technique, to fabricate mucoadhesive films for oral mucosal drug delivery. The designed printing formulation includes PEGDA as the printing resin, TPO as the photoinitiator, tartrazine as the photoabsorber, PEG 300 as the additive and HPMC as the bioadhesive material. The influence of printing formulation and printing parameters on the printing formability of the oral films were elucidated in depth, and the results suggested that PEG 300 in the formulation not only provided the necessary flexibility of the printed oral films, but also improved drug release rate due to its role as pore former in the produced films. The presence of HPMC could greatly improve the adhesiveness of the 3D-printed oral films, but excessive HPMC increased the viscosity of the printing resin solution, which could strongly hinder the photo-crosslinking reaction and reduce printability. Based on the optimized printing formulation and printing parameters, the bilayer oral films containing a backing layer and an adhesive layer were successfully printed with stable dimensions, adequate mechanical properties, strong adhesion ability, desirable drug release and efficient in vivo therapeutic efficacy. All these results indicated that an LCD-based 3D printing technique is a promising alternative to precisely fabricate oral films for personalized medicine.

## 1. Introduction

In the past decades, oral mucoadhesive film has emerged as an attractive mucosal drug delivery platform owing to its distinct properties such as convenient administration, rapid onset and high bioavailability [1]. It is designed to adhere to the oral mucosa and release the drug controllably [2] so as to achieve the therapeutic effect locally or systemically. Notably, by this way, the drug is absorbed by the mucosa and directly enters into the blood stream, which can avoid first-pass effects both in the gastrointestinal tract and in the liver. In addition, with the advantages of convenience, fixed absorption area and better compliance, the oral films are especially suitable for children with poor medication compliance and the elderly with dysphagia [3].

Marketed oral films are currently manufactured by solvent casting [1,4]. Basically, drugs, film-forming polymer excipients and other additives ared dissolved or dispersed into a suitable solvent to form a solution or suspension; then, the uniform solution is poured into a predesigned mold and followed by a drying process to obtain the required product [3]. As the most widely used fabrication method for oral films, solvent casting has noteworthy advantages such as adequate uniformity, scalable capacity of production and accurate dose [5]. However, there still remains some ongoing limitations. Initially, the use of organic solvent causes high risk of residue and raises security and environment-related concerns, which greatly limits its extended applications [6]. Additionally, it is extremely difficult to dry the film produced by the solvent casting, which needs to be carefully balanced to avoid both excessive drying and insufficient drying. Most importantly, solvent casting is ineffective in producing customized oral films with complex structures with high precision. As a result, there have recently been many new fabrication approaches developed to prepare oral films, including hot-melt extrusion [7,8], electrospinning [9,10,11] and electrostatic spray deposition [12,13]. Despite avoiding the use of organic solvent in these techniques, the temperature variations, complexity and cost-added procedures of these techniques also hinder their practical industrialization and commercialization in manufacturing oral films.

A 3D printing technique [14,15] is capable of selectively producing various products with specific structures through layer-by-layer printing and curing [16]. In the past decades, numerous dosages such as tablets [17,18], pellets [19], transdermal microneedles [20], hydrogels [21] and oral films [22,23] have been produced using this technology. Fatouros et al. [24] used fused deposition modeling (FDM) 3D printing technology to prepare PVA-based oral adhesive films. Compared with previously mentioned methods, FDM-printed oral film has better structural characteristics and dose uniformity. Chitosan was incorporated into polymer filaments to enhance adhesion and permeability of the buccal film. However, FDM requires temperature rise during processing, which will lead to potential degradation of heat-sensitive drugs. Furthermore, low printing accuracy and high roughness of the product surface printed by FDM strongly retard its further applications, particularly in producing oral films with high precision. Liquid crystal display (LCD) [25,26] is one of the photopolymerization-based 3D printing techniques [27], which could irradiate liquid resin to form a 3D entity, with ultraviolet light-emitting diodes as the light source and liquid crystal display as the imaging system. Notably, LCD printer equipment is small in size, which meets the development trend of desktop 3D printing. More importantly, LCD is suitable for most photosensitive resin materials and could be extensively applied to print various products with highly precise structures.

The present study aims to utilize the LCD technique to prepare mucoadhesive bilayer oral films for oral mucosal drug delivery, to perform an in-depth investigation on the influence of the printing formulation and printing parameters on the printability of oral films, and to fully evaluate their characteristics including mechanical properties, swelling ability, adhesion ability, stability, drug release and in vivo therapeutic efficacy.

## 2. Materials and Methods

### 2.1. Materials

Polyethylene (glycol) Diacrylate (PEGDA, average MW 170.163) was provided by German Fine Chemical Ltd. HPMC-E5/E15/K4M/K100M was provided by Anhui Shanhe Pharmaceutical Co., Ltd. (Anhui, China). Eudragit^®^ RL 100 was provided by Ashland Chemical Trading Co., Ltd. (Shanghai, China). Polyethylene glycol (PEG 200/300/400), sodium alginate (average MW 198.11), poly (vinyl alcohol) (PVA, average MW 402.58), tartrazine (average MW 316.353), sodium chloride (average MW 58.44), methanol (average MW 32.04), sodium alginate (average MW 198.11), acetonitrile (average MW 41.052), chloral hydrate (average MW 165.4) and saline solution were purchased from Sinopharm Reagent Co., Ltd (Shanghai, China). Sodium carboxymethyl cellulose (CMC-Na, average MW thousands to millions), sodium dodecyl sulfate (average MW 288.379), dexamethasone acetate (average MW 434.498), diphenyl (2,4,6-trimethylbenzoyl) phosphine oxide (TPO), chitosan (CHI), hydroxypropyl cellulose(HPC), Hydroxypropyl Methyl Cellulose (HPMC, average MW 243.96n) were purchased from Aladdin Chemicals Ltd. (Shanghai, China). Dexamethasone acetate oral mucoadhesive patches (Yiketie^®^) were purchased from Shenzhen Taitai Pharmaceutical Co., Ltd. (Shenzhen, China).

### 2.2. Preparation of the Printing Formulations

The printing formulations were prepared with different amounts of PEGDA, PEG 300, TPO and tartrazine according to the following tables (Table 1, Table 2, Table 3 and Table 4). PEG 300 was firstly weighed, then the photoinitiator (TPO) was added, followed by an addition of a certain amount of photopolymerization monomer PEGDA, which was fully mixed, heated in a light-resistant condition and stirred until completely dissolved. Table 1 provides the predesigned printing formulations to investigate the influence of PEG 300 and PEGDA on the printability of the oral films. Table 2 provides the predesigned printing formulations to investigate the influence of the photoinitiator (TPO) on the printability of the oral films. Table 3 provides the predesigned printing formulations to investigate the influence of the photoabsorber (tartrazine) on the printability of the oral films. Table 4 provides the predesigned printing formulations to investigate the influence of the bioadhesive material (HPMC) on the printability of the oral films. Adding the preparation process of this formulation: HPMC was weighed, then the certain amounts of PEG, TPO and PEGDA were weighed and added, fully mixed and stirred to prepare the HPMC resin solution with concentrations of 0%, 10%, 20% and 30%.

### 2.3. 3D Printing of Oral Films

Oral films and specimens for mechanical tests were firstly designed by the Solidworks 2018 software (Figure 1). The dimension of the specimen was according to the GB/T 1040.1-2018. The stereolithography (STL) file was generated with the assistance of Photon_WorkShop_V2.1.24 slicing software. Then, the STL was imported into the Anycubic Mono X 3D printer. Specimens (specimens for mechanical tests) and oral film samples were printed in the different concentrations mentioned in Section 2.2 of resin with a layer height of 20 μm and a layer-by-layer exposure time of 4 s as the printing parameters.

Three models of oral films were designed by Solidworks 2018 software [28] (solid sample, hollow mesh, concentric circle hole), as shown in Figure 2, with a diameter of 10 mm and a thickness of 300 μm. Three model entities were prepared based on the Section 2.2 formulations of printing. Under the condition that the exposure time of each layer (3 s) was constant, three different models were printed one by one with adjusting different layer heights (20, 30, 50 μm). Then, under the fixed layer height (20 μm), the exposure time of each layer was adjusted (3, 4, 5 and 6 s). Table 5 provides the predesigned printing parameters to investigate the influence of layer height and layer-by-layer exposure time on printing formability.

### 2.4. Appearance Characterization of the 3D-Printed Oral Films

A total of 15 pieces of film samples with different formulations were randomly selected. Then, an electronic vernier caliper [29] was used to measure the edge and center of the film at 5 different position points, the reading was recorded, followed by the organization and collation of the diameter and thickness data.

### 2.5. Mechanical Properties 

The breaking force was measured by the Texture Analyser (TA. XT Plus, SMS, UK), and the tensile strength and elongation at break of specimens with different formulations were calculated. With the assistance of the Texture Analyser, the specimen was fixed using a tensile test fixture and stretched at a constant rate of 0.5 mm/s, with a trigger force of 0.049 N and a tensile distance of 5 mm to measure its breaking force. The formulas were shown in the following:$$\sigma =\frac{F}{D\times T}$$
$$\varepsilon =\frac{\Delta L}{L}$$

σ—Tensile strength, MPa;

F—Breaking force, N;

D—The diameter of the film, mm;

T—The thickness of the film, mm;

Ε—Elongation at break, %;

ΔL—The difference value between the length at break and the original length, mm;

L—Original length, mm.

### 2.6. Mucoadhesive Ability Tests 

The adhesion of the film was tested using the Texture Analyser (TA. XT Plus, SMS, UK). Fresh egg shell membrane was taken and washed with 0.9% normal saline. Then, 1 cm^2^ fresh egg film was fetched and glued to the P6 probe with double-sided adhesive. Next, the probe was moved to a fixed position. The oral film was added by 50 μL of artificial saliva dropwise to be fully infiltrated; then, it was pressed 30 s under compression mode, where the speed before the test was 0.5 mm/s, and the test speed was 0.5 mm/s, with the trigger force of 0.049 N. After that, in tensile mode and at the same test speed, the force measured when the film was separated from the egg shell membrane was exactly the adhesion force. The films with different HPMC contents were tested according to the above mode 9 times each. 

### 2.7. 3D Printing of the Bilayer Oral Films

With the assistance of Solidworks2018 software, 8 mm, 10 mm and 12 mm diameter hollow grid bilayer films were designed and imported into Photon_WorkShop_V2.1.24 slicing software to generate STL slice files with a layer height of 20 μm, a layer-by-layer exposure time of 5 s, and a total of 15 layers.

Initially, photoinitiator TPO and photoabsorber (tartrazine) were accurately weighed and put into a beaker containing a certain amount of PEGDA; then, 1% (*w*/*w*) HPMC K4M that had completely swollen under light-resistant conditions was added, shaken, sonicated and stirred until completely dissolved. After that, PEG300 containing API (dexamethasone acetate) was added to form a resin solution of adhesion layer containing the drug. Then, TPO was precisely weighed and put into a beaker containing a certain amount of PEGDA, followed by an addition of PEG300 in the dark, mixed thoroughly until completely dissolved as a resin solution for the backing layer. Table 6 provides the concrete predesigned printing formulations of the bilayer oral films.

Then, the STL slice file was imported into the LCD printer. Firstly, the drug-containing adhesive layer resin tank was loaded, then the drug-containing layer was printed. Next, the former tank was replaced by the backing layer resin tank to continue printing the backing layer. After finishing, the samples were taken out and the residual resin was removed using filter paper, stored at room temperature and protected from light and cold with plastic sealing tape.

### 2.8. Characterization of the 3D-Printed Bilayer oral Films

#### 2.8.1. Appearance

The appearance of the 3D-printed bilayer oral films was investigated according to the methods in Section 2.4. For the thickness and diameter, the 10 pieces of film of different batches and different diameters were randomly selected. For the weight difference, 20 pieces for each batch of bilayer films were randomly selected.

As for the weight difference, 20 pieces each of the different formulations of film were accurately weighed by an electronic balance (AB135-S, Mettler Toledo), and both the individual and the average weight were achieved. 

#### 2.8.2. Mechanical Properties

The breaking force of the 3D-printed bilayer oral films with different diameters (10, 12 mm) was measured by the Texture Analyser (TA. XT Plus, SMS, UK) according to the methods in Section 2.5.

#### 2.8.3. Bioadhesive Ability

The bioadhesive ability of the 3D-printed bilayer oral films was investigated according to the methods in Section 2.6. The 3D-printed bilayer films were tested according to these methods 6 times each. The average value was calculated. Referring to the small cup paddle method to measure the adhesion time of oral film, 1 cm^2^ of fresh egg film was taken and pasted on the stirring paddle of the dissolution instrument with double-sided tape. Then, 50 μL of artificial saliva dropwise was added to the egg film and pressed for 1 min. After that, we added 100 mL of artificial saliva at 30 r/min at 37 °C to record the time of film shedding.

#### 2.8.4. Scanning Electron Microscopy (SEM)

Scanning electron microscope (SEM) was used to observe the microstructure and surface roughness of the 3D-printed bilayer oral films. The bilayer oral film was dried under vacuum at room temperature for 24 h. One of them was taken to be cut into two from the middle and was fixed and treated with gold spray using conductive glue. Then, the morphology of the bilayer film was observed under a scanning electron microscope (SEM, Hitachi S-4700 Hitachi High-Tech Science Corporation, Tokyo, Japan).

#### 2.8.5. Physical State of the API and Printed Films by DSC and XRD

DSC: The API, blank films and drug-loaded bilayer films were analyzed by differential scanning calorimetry (DSC1/700, Mettler-Toledo, Zurich, Switzerland). About 10 mg of powder samples were weighed and put into aluminum pans, then heated from 30 to 260 °C at a heating rate of 10 °C/min under a nitrogen atmosphere.

XRD: The physical structure (crystal or amorphism) API, blank resin film and drug-containing film samples were determined and compared by the X’Pert PRO (PNAlytical, Netherlands) using a Cu Ka X-ray source (l = 1.5418 Å). The samples were analyzed in a 10–80°2θ range at 40 KV and 40 mA to record the diffraction peak and plot the XRD map.

#### 2.8.6. Drug Content

A total of 6 pieces of the 3D-printed oral films for each sample were taken and grinded with a mortar. Then, an appropriate amount was accurately weighed and put in a 100 mL beaker, followed by an addition of 60 mL of 0.35% sodium lauryl sulfate to each sample, which was sonicated and shaken to completely dissolve. Then, a 0.45 μm microporous filter membrane was used to filter the achieved solution. Under the chromatographic conditions, where the acetonitrile–water (60:40) was used as a mobile phase, the drug content was determined with high-performance liquid chromatography (HPLC, ThermoFisher U3000) at 241 nm based on the valid calibration curve of dexamethasone acetate (A = 0.6645C−0.07531, R = 0.9999).

#### 2.8.7. In Vitro Drug Release Tests

Drug release from the 3D-printed bilayer oral films were investigated with a USP-II apparatus (Apparatus 2, paddle; Tianda Tianfa Technology Co., Ltd., Tianjin, China). Samples of 3D-printed bilayer oral films containing 2% and 5% drug (dexamethasone acetate) and commercially available dexamethasone acetate oral films were detected by HPLC according to the methods in Section 2.8.8 at the following sampling points (10, 20, 40, 60, 90, 120 and 180 min). For every sampling, 5 mL of medium was taken out from the samples and replaced by 5 mL of fresh rehydration solution to keep the drug releasing in a sink state. 

#### 2.8.8. In Vivo Pharmacodynamics

After 5 days of adaptive feeding, 20 SD rats were randomly divided into 4 groups (blank group, model group, positive group and DEX group) with 5 rats in each group. The rats were anesthetized by chloral hydrate (300 mg/kg) before modeling. An amount of 0.1 mL of 30% glacial acetic acid solution was injected into the oral mucosa on the lower left side of the rat. Then, the injection site was wiped using skim cotton soaked with saline. The gray-white blisters were formed locally. After that, ulcer lesions were observed after 24 h.

After successful modeling, the positive group and the DEX group were treated with drugs. Before administration, rats were injected with 10% chloral hydrate anesthesia. Then, the film was attached to the ulcer area by gently pressing for 20 s to ensure a tight fit with the ulcer. Each administration time was at least 30 min. The blank group and the model group were given normal feeding without any drug treatments. In the positive group, commercially available dexamethasone acetate films were given to the ulcer with a labeled amount of 0.3 mg, half of which was administered once a day. The DEX group was given an 8 mm diameter bilayer film at a dose of 0.15 mg each, administered once a day for 7 days. Food and water intake were forbidden within 2 h after administration to ensure the efficacy of the drug.

The pictures of the ulcer surface were taken using a camera (Canon 200d) with a fixed position (same height from the ulcer surface) and a ruler (BAOKE-RU2074). The obtained pictures and the ulcer areas were analyzed and calculated by Image J software.

After 7 days of administration, rats in the blank group, model group, positive group and DEX group were anesthetized with excessive chloral hydrate intraperitoneally injected to euthanize them. Then, the ulcer surface tissue was cut out and rinsed with 0.9% sodium chloride solution. After that, 4% paraformaldehyde was used to fix for 48 h. The ulcer surface tissue was taken out to make oral ulcer sections after dehydration and paraffin embedding. After hematoxy-eosin staining and being dehydrated and sealed, tissue changes were observed under the optical microscope.

### 2.9. Statistical Analysis

All data were presented as mean ± standard deviation (SD) values. The statistical analyses for oral ulcers were performed using Student’s *t*-test (*p* < 0.05).

## 3. Results and Discussion

### 3.1. Influence of the Printing Formulation on the Printability

#### 3.1.1. Effect of Monomer and Pore Formers on Printing Formability

Practically, PEG 300 was utilized in the printing formulation as a pore former to enhance drug release from the 3D-printed oral films. However, its presence may have a negative influence on the mechanical properties and printability of the achieved oral films. As shown in Figure 3A, as the PEG concentration increased from 0% to 40% (further increasing of PEG content caused printing failure), the breaking force of the specimens decreased gradually, indicating that the presence of PEG in the printing formulation reduced the mechanical properties of the 3D-printed specimen. In addition, the dimensions (diameter, thickness, variations of diameter and thickness) of the 3D-printed oral films were also strongly impacted by the presence of PEG (Figure 3B–D). Furthermore, more PEG in the formulation caused a larger deviation from the predesigned dimensions. This is because PEG was uniformly dispersed in the pore structure of the monomer (PEGDA) molecule, which affected the photo-crosslinking reaction process, resulting in nonuniform solidification of the printed body. 

#### 3.1.2. Effect of Photoinitiator on Printability

During a photocuring reaction, the photoinitiator initiates the polymerization of the free radicals produced by the monomer under ultraviolet light, which causes cross-linking and solidification. So, it is necessary to carefully balance the content of the photoinitiator in the printing formulation to avoid both excessive polymerization and insufficient cross-linking. As shown in Figure 4A, an increase in TPO led to an enhancement of the break force of the achieved specimen, indicating better mechanical properties of the obtained product. For the 3D-printed oral films, when the TPO concentration was less than 0.6%, the printed oral films were incomplete (Figure 4B), and the diameter deviated from the standard value more largely. When increasing the content of TPO to 0.8%, the printing formability was adequately good, and the dimensions of the achieved oral films were closer to the standard size. 

#### 3.1.3. Effect of Photoabsorber on Printability

Practically, photoabsorber is commonly applied in the printing formulation to avoid undesirable polymerization and to reduce over-curing during the printing process, dramatically promoting the printability of the corresponding products. In the present study, the influence of a commonly used photoabsorber, tartrazine, on the printability of the oral films was evaluated and the results were shown in Figure 5. With the increase in content of the tartrazine from 0 to 0.02%, the breaking force of the printed specimens increased but then slightly decreased when the tartrazine content was further increased to 0.03% (Figure 5A). For the 3D-printed oral films, 0.01% and 0.02% of tartrazine provided sufficient printability with stable and consistent dimensions (diameter and thickness and their variations, (Figure 5B–D)). This is because the photoabsorber could weaken the photo-crosslinking reaction and improve the printing accuracy. If the concentration was too high, the photo-crosslinking reaction was insufficient, resulting in poor integrity of the print body. 

#### 3.1.4. Effect of Adhesive Material on the Printability and Adhesive Ability

Commonly used bioadhesive materials, including sodium alginate (SA), chitosan (CHI), sodium carboxymethyl cellulose (CMC-Na), hydroxypropyl cellulose (HPC), poly (vinyl alcohol) (PVA) and HPMC (E5/E15/K4M/K100M), were applied and mixed into the printing resin (PEGDA), as shown in Figure 6A. Except HPMC (E5/E15/K4M), it was impossible for other materials to fully and completely dissolve in the resin, which would strongly hinder the printing process and formation of an acceptable printing product. As a result, HPMC (E5/E15/K4M) was selected in the further investigations. Amongst these, K4M provided the lowest viscosity (Figure 6B) when included in the printing resin with 10% over two other types of HPMC; hence, K4M was utilized in the following investigation. As shown in Figure 6C,D, more K4M not only reduced the mechanical properties of the printed specimens, but also enlarged the deviations from the predesigned dimensions (diameter and thickness), indicating that the presence of K4M strongly reduced the printing accuracy of the films. However, as shown in Figure 6E, more K4M in the printing formulation brought stronger mucoadhesive force, which is desirable when the 3D-printed films were administered in the oral mucosal. In a word, when using HPMC K4M as a bioadhesive material, its content needs to be carefully balanced in the printing formulation.

### 3.2. The Influence of the Printing Parameters on the Printability

Besides the printing formulation, processing parameters, especially the layer height and exposure time, could also have a big impact on the printability of the printed oral films. As shown in Figure 7, a more complete printing film could be achieved either by applying thinner layer height, or by conducting a longer exposure time. These results could be confirmed by Figure 8; the 10 μm layer height printing films were complete, but the thickness of the printed body deviated to be larger than the standard size (Figure 8 A,C). When the layer height was increased to 50 μm, part of the printed body was incomplete and unable to be cured. Taking the printing speed and accuracy into consideration, 20 μm was selected as the optimized printing height. Additionally, the falling off and incomplete printing body might also be related to the insufficient exposure time. However, when the exposure time was increased to 6 s, the hollow grid structure was blurred (Figure 7) and the printability was poor (Figure 8D–F). In order to guarantee an adequate printability and structure completion, the exposure time was optimized as 5 s.

### 3.3. Characterization of 3D-Printed Bilayer Oral Films

#### 3.3.1. Appearance Characteristics

The thickness, diameter and weight of different batches of 3D-printed bilayer films were recorded and the results were shown in Table 7 and Figure 9. The appearance of the 3D-printed bilayer oral films was complete, with a smooth surface and integrated edges. The weight difference of the achieved films was less than 10%. The thickness of the printed films was about 0.32 mm, which was less than 10% difference over the theoretical size (0.3mm). During printing, the resin tank needed to be replaced between the bilayer film printing process. As a result, after the printing of the drug-containing adhesion layer, there was still residual resin on the surface of the sample, which was cured together using ultraviolet rays with the backing layer resin, resulting in an increased thickness. All the real diameters of the bilayer films with a predesigned size of 8, 10 and 12 mm were increased by 2–3%, which could be explained by the increased exposure intensity within the part closed to the platform, resulting in an excessive solidification of the solid edge with an increased diameter.

#### 3.3.2. Mechanical Properties and Bioadhesive Ability 

The breaking force of the 3D-printed oral films was further investigated, and the results are shown in Figure 10. Firstly, the size of the 3D-printed oral films did not have a big impact on the breaking force thereof; however, the thickness of the backing layer did. As shown in Figure 10, the breaking force of the single-layer film was about 0.5 N, which was dramatically increased with the presence of the backing layer. Additionally, a thicker backing layer resulted in a stronger breaking force of the achieved oral films. However, if the backing layer is too thick, it would increase the sensation of foreign films in the mouth and reduce the patient’s compliance. 

Additionally, the 3D-printed bilayer oral films possessed adequate adhesive force (Table 8) with acceptable adhesive time, indicating that 3D-printed bilayer oral films have sufficient bioadhesive ability.

#### 3.3.3. Morphology and Physical State

The cross-sectional morphology of the 3D-printed oral films was investigated, and the result was shown in Figure 11A. The upper part was the drug-containing adhesion layer, whose surface structure is relatively rough with porous structures, which is conducive to the adhesion of the oral mucosa and to achieve rapid drug release. The lower part was the backing layer, whose surface shape was smooth, with a dense and compact structure. The backing layer contained higher concentration of PEGDA, resulting in a denser structure, which could hinder the leakage of drugs from the backing layer and allow a desirable unidirectional drug release.

The physical state both of the API itself and the drug in the printed oral films was investigated with DSC (Figure 11B) and XRD (Figure 11C). Results from both DSC and XRD revealed that the API (dexamethasone acetate) was clearly in a crystal state and turned into an amorphous state after being printed with the photocurable resin. This transition of the physical state of API is quite auspicious because a drug in an amorphous state would dissolve much faster than in a crystal state, which is capable of allowing a prompt drug release and quick drug absorption of the 3D-printed oral films when being administered.

#### 3.3.4. Drug Content and In Vitro Drug Release

Drug content of the 3D-printed bilayer oral films was investigated to evaluate the potential drug loss during the 3D printing process. As shown in Figure 12A, drug contents from all three types of oral films were around 90% (89.1, 88.9 and 90.1 for the three groups), indicating that there was no significant loss during the printing process. This could be a potential benefit of the LCD printing technique compared with FDM printing, which involves a dramatic temperature fluctuation and might cause significant drug degradation [19,30].

In vitro drug release rate is also a critical quality for oral films. As shown in Figure 12B, 3D-printed oral films produced a faster drug release over the commercial one, and it could be further manipulated by adjusting the PEG content in the printing formulation. However, when increasing the drug ration from 2% to 5% in the printing formulation, drug release rate significantly dropped. This might be caused by the low solution of the drug in the printing resin (PEGDA); the surplus drug precipitated as crystal instead of an amorphous state and dramatically reduced the drug release rate [31,32]. 

### 3.4. In Vivo Pharmacodynamics Studies

The 3D-printed bilayer oral films were administered in an ulcer-bearing rat model, and the results after treatments were shown in Figure 13. Compared with the model group, the area of oral ulcers (*p* < 0.05) was significantly reduced both in the DEX group and in the positive group (commercial film). After five days of treatments, the area of oral ulcers in the DEX group was even lower than in the positive group (Figure 13A), indicating that the 3D-printed oral films had a sufficient therapeutic effect. This was further confirmed by the result of the HE staining. As shown in Figure 13B, the epithelial tissue structure of the normal group was complete and clear while the epithelial structure of the oral mucosa in the model group was destroyed. A large number of inflammatory cells were distributed in the basal layer, suggesting a high degree of inflammation. Compared with the model group, the density of distributed neutrophils and the degree of inflammatory cell infiltration were significantly reduced in the DEX group and the positive group. In addition, the degree of inflammatory cell infiltration in the DEX group was lower than that in the positive group, indicating that the 3D-printed bilayer films had a better therapeutic effect on the oral ulcer inflammation over the commercial films.

## 4. Conclusions

In this study, the LCD-based 3D printing technique was successfully applied to produce bilayer films for oral mucosal drug delivery. A 3D-printable formulation was developed and optimized with PEGDA as the printing resin, TPO as the photoinitiator, tartrazine as the photoabsorber, PEG 300 as the additive and HPMC as the bioadhesive material. The properties of the printed oral films, including the appearance, printability, mechanical properties, bioadhesive ability and drug release rate could be manipulated by adjusting the formulation ratio of the above materials, especially the PEG 300 and HPMC. PEG 300 in the formulation not only provided the necessary flexibility of the printed oral films, but also improved the drug release rate due to its role as pore former. The presence of HPMC could greatly improve the adhesiveness of the 3D-printed oral films, but excessive HPMC increased the viscosity of the printing resin solution, which could strongly affect the photo-crosslinking reaction and reduce the printability. Additionally, the printing parameters including the printing layer height and the exposure time also had a big impact on the printability of the oral films. Additionally, there was a transition of the drug physical state from a crystal to an amorphous state after the 3D printing process, which also promoted the drug release rate and in vivo absorption. The in vivo studies confirmed that the 3D-printed bilayer oral films are qualified for the treatment of ulcers with oral mucosal administration. All the results suggested that the LCD technique is a promising technique to print oral films for oral mucosal drug delivery, particularly the customized ones for personalized medicine.

## Figures and Tables

**Figure 1 pharmaceutics-15-00758-f001:**
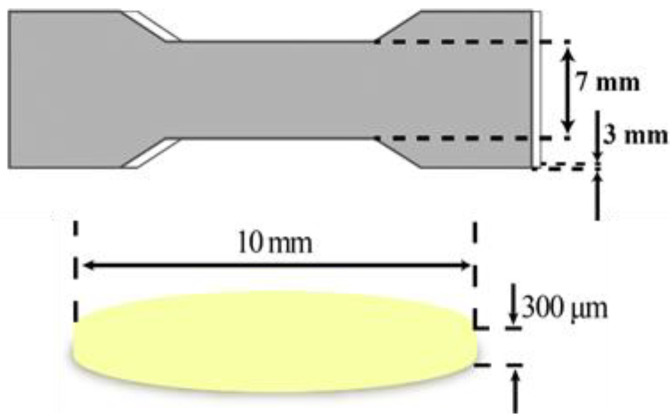
The dimensions of specimen for mechanical tests and oral film samples.

**Figure 2 pharmaceutics-15-00758-f002:**
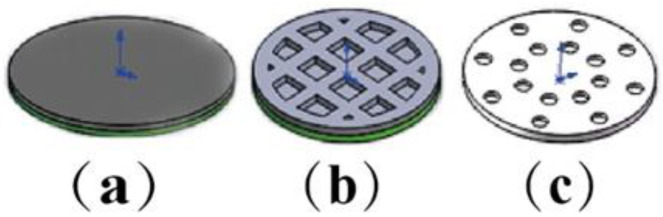
The models of oral films: (**a**) solid sample, (**b**) hollow grid, (**c**) concentric holes.

**Figure 3 pharmaceutics-15-00758-f003:**
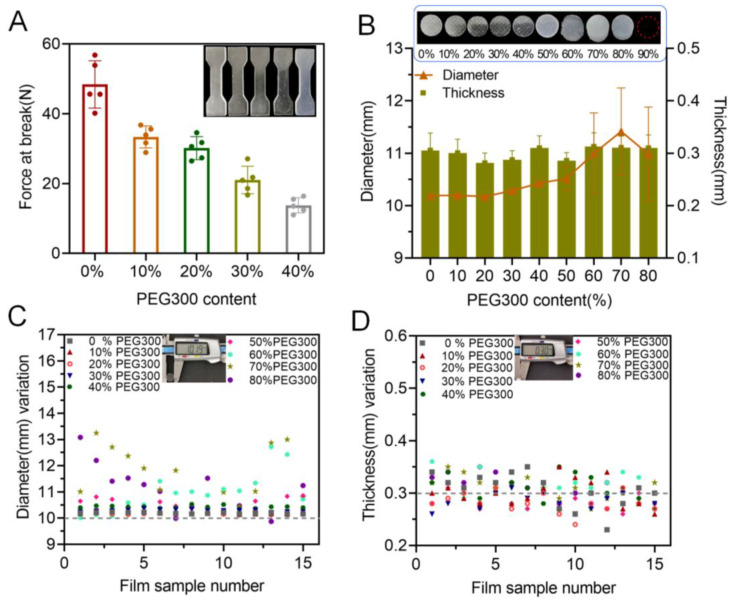
Effect of PEG concentration on printability: (**A**) effect of PEG content on the tensile strength (n = 6); (**B**) effect of PEG content on the diameter and thickness of the produced oral films (n = 15); (**C**) diameter variations of the 3D-printed oral films; (**D**) thickness variations of the 3D-printed oral films.

**Figure 4 pharmaceutics-15-00758-f004:**
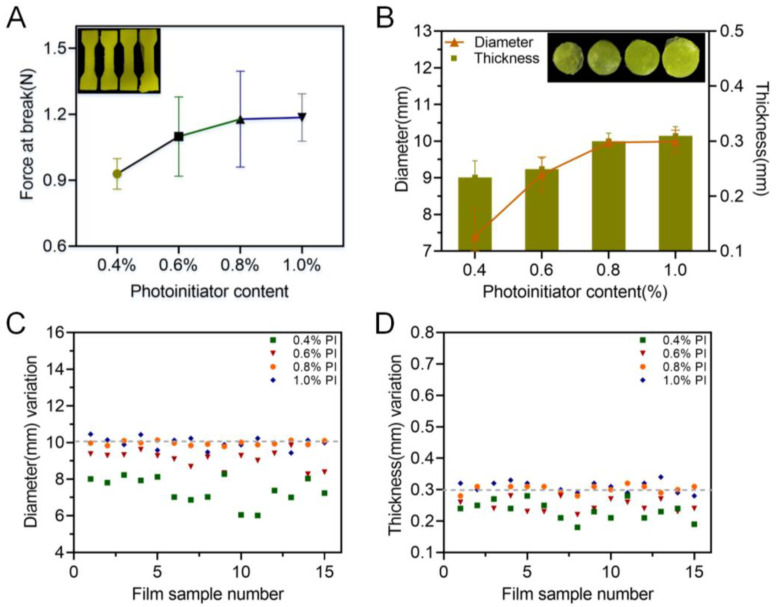
Effect of the photoinitiator on the printability: (**A**) force at break of specimen samples (n = 6); (**B**) thickness and diameter of 3D-printed oral film samples (n = 15); (**C**) diameter variations and (**D**) thickness variations of 3D-printed oral film samples.

**Figure 5 pharmaceutics-15-00758-f005:**
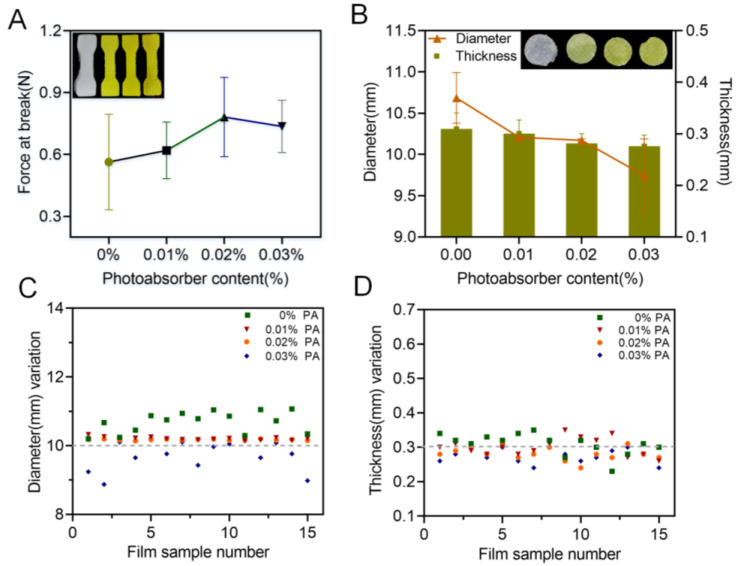
Effect of the photoabsorber on the printability: (**A**) force at break of specimen samples (n = 6); (**B**) thickness and diameter of 3D-printed oral film samples (n = 15); (**C**) diameter variations and (**D**) thickness variations of 3D-printed oral film samples.

**Figure 6 pharmaceutics-15-00758-f006:**
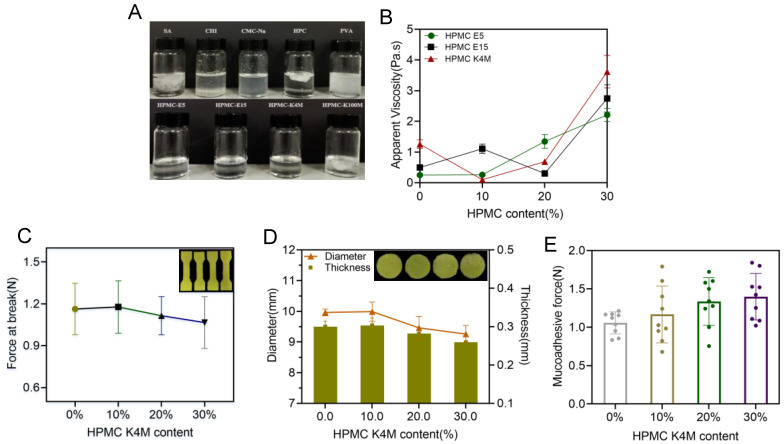
(**A**) The physical state of adhesive materials mixture; (**B**) viscosity of the printing resin containing HPMC E5, E15, K4M at different concentrations (n = 6); (**C**) effect of HPMC K4M on the force at break of 3D-printed oral films (n = 6); (**D**) effect of HPMC K4M on the thickness and diameter of 3D-printed oral films(n = 15); (**E**) effect of HPMC K4M on the mucoadhesive force of 3D-printed oral films (n = 6).

**Figure 7 pharmaceutics-15-00758-f007:**
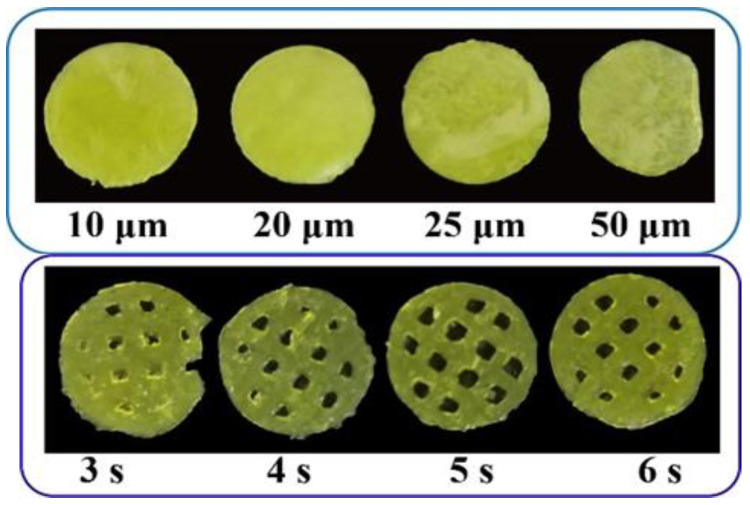
Pictures of 3D-printed oral films with different printing parameters.

**Figure 8 pharmaceutics-15-00758-f008:**
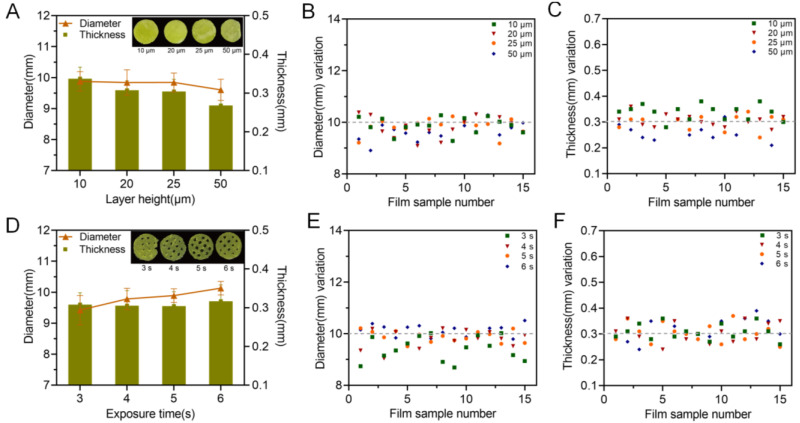
Influence of printing parameters on the printability of the oral films. Exposure time and layer height; (**A**) effect of layer height on the diameter and thickness of 3D-printed oral films (n = 15); (**B**) diameter variations and (**C**) thickness variations of 3D-printed oral film samples from (**A**); (**D**) effect of exposure time on the diameter and thickness of 3D-printed oral films (n = 15); (**E**) diameter variations and (**F**) thickness variations of 3D-printed oral film samples from (**D**).

**Figure 9 pharmaceutics-15-00758-f009:**
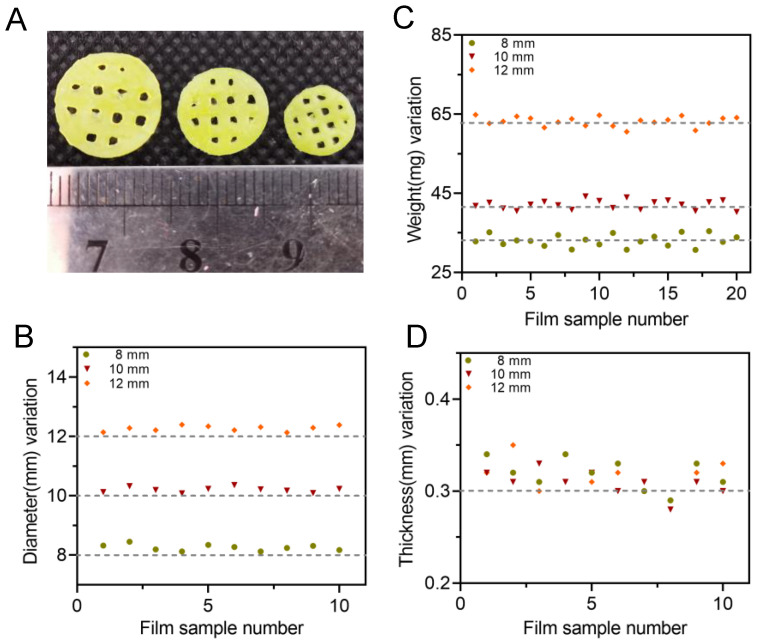
(**A**) Appearances of 3D-printed oral films with three different diameters; (**B**) diameter variation of oral films (n = 10); (**C**) weight variation of oral film (n = 20) and (**D**) thickness of oral films (n = 10).

**Figure 10 pharmaceutics-15-00758-f010:**
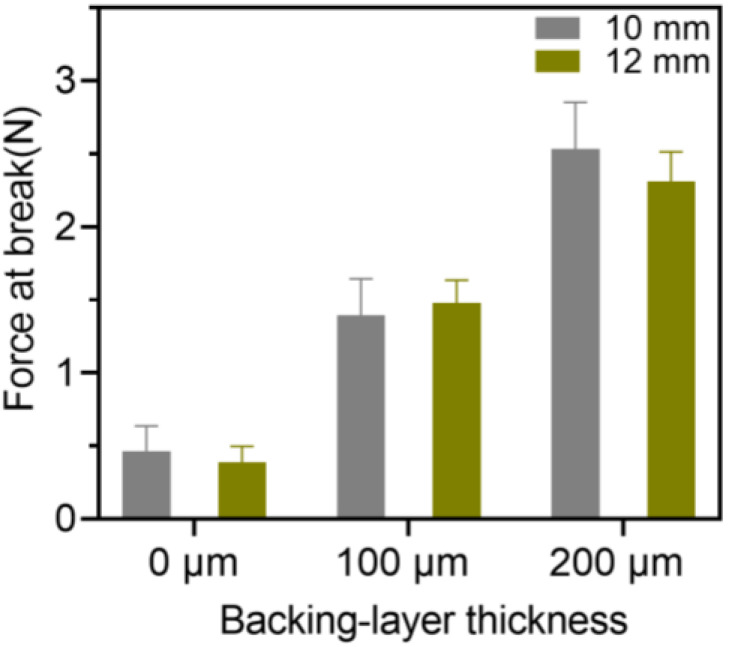
Breaking force of the 3D-printed bilayer oral films (n = 6).

**Figure 11 pharmaceutics-15-00758-f011:**
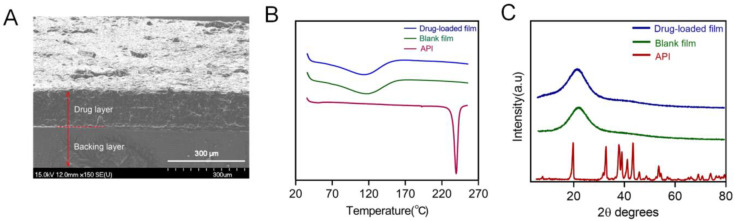
(**A**) Cross-sectional morphology of the printed bilayer film; (**B**) DSC analysis pattern spectra of dexamethasone acetate, polymer film, drug-loaded bilayer film; (**C**) the X-ray Powder Diffraction spectra of dexamethasone acetate, polymer, drug-loaded bilayer film.

**Figure 12 pharmaceutics-15-00758-f012:**
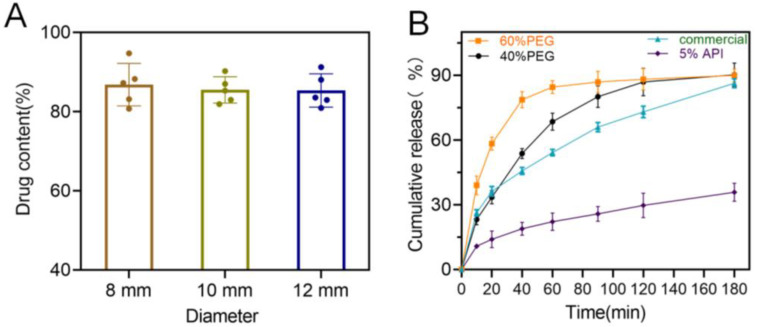
(**A**) Drug content of the 3D-printed bilayer oral films (n = 6); (**B**) in vitro drug cumulative release from the 3D-printed bilayer oral films.

**Figure 13 pharmaceutics-15-00758-f013:**
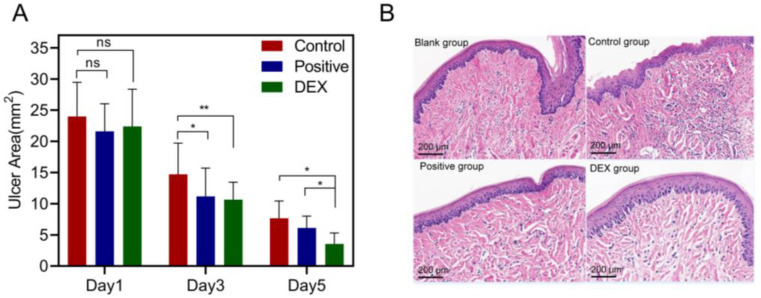
(**A**) Ulcer area changes of the model, positive and DEX groups after treatments (n = 4, * *p* < 0.05, ** *p* < 0.01, ns: no significant difference); (**B**) HE staining results of different groups after treatments.

**Table 1 pharmaceutics-15-00758-t001:** The printing formulations with different ratios of PEG 300 and PEGDA.

PEG 300 (g)	PEGDA (g)	TPO (g)
0	99.00	1.00
10.00	89.00	1.00
20.00	79.00	1.00
30.00	69.00	1.00
40.00	59.00	1.00
50.00	49.00	1.00
60.00	39.00	1.00
70.00	29.00	1.00
80.00	19.00	1.00
90.00	9.00	1.00

**Table 2 pharmaceutics-15-00758-t002:** The printing formulations with different ratios of photoinitiator (TPO).

PEG300 (g)	PEGDA (g)	Tartrazine (g)	TPO (g)
80.00	19.58	0.02	0.40
80.00	19.38	0.02	0.60
80.00	19.18	0.02	0.80
80.00	18.98	0.02	1.00

**Table 3 pharmaceutics-15-00758-t003:** The printing formulations with different ratios of photoabsorber (tartrazine).

PEG300 (g)	PEGDA (g)	TPO (g)	Tartrazine (g)
80.00	19.00	1.00	0
80.00	18.99	1.00	0.01
80.00	18.98	1.00	0.02
80.00	18.97	1.00	0.03

**Table 4 pharmaceutics-15-00758-t004:** The printing formulations with different ratios of bioadhesive material.

HPMC E5/E15/K4M (g)	PEG300 (g)	PEGDA (g)	Tartrazine (g)	TPO (g)
0	79.18	20.00	0.02	0.80
10.00	69.18	20.00	0.02	0.80
20.00	59.18	20.00	0.02	0.80
30.00	49.18	20.00	0.02	0.80

**Table 5 pharmaceutics-15-00758-t005:** The parameters of 3D printing.

Exposure Time of Each Layer (s)	Layer Height (μm)
3	20
3	30
3	50
4	20
5	20
6	20

**Table 6 pharmaceutics-15-00758-t006:** Printing formulation of the bilayer oral films.

	API	PEGDA	PEG300	HPMC K4M	TPO	Tartrazine
Adhesion layer containing drug	2.00 g	20.00 g	59.18 g	20.00 g	0.80 g	0.02 g
Backing layer		69.2 g	30.00 g	/	0.80 g	/

**Table 7 pharmaceutics-15-00758-t007:** The thickness, diameter, weight of different batches of bilayer films.

Theoretical Diameter (mm)	Thickness (mm, n = 10)	Diameter (mm, n = 10)	Weight (mg, n = 20)
8	0.32 ± 0.01	8.25 ± 0.11	33.03 ± 1.53
10	0.31 ± 0.02	10.20 ± 0.10	42.11 ± 1.17
12	0.32 ± 0.02	12.27 ± 0.09	64.14 ± 1.24

**Table 8 pharmaceutics-15-00758-t008:** Bioadhesive ability of 3D-printed oral films with different diameters (n = 6).

Diameter (mm)	Adhesion Force (N)	Adhesion Time (h)
8	1.23 ± 0.51	0.53 ± 0.19
10	0.98 ± 0.38	0.62 ± 0.23
12	1.27 ± 0.45	0.57 ± 0.14

## Data Availability

The data presented in this study are available on request from the corresponding author.

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
