# Peer review of "Three-Dimensional-Printed Oral Films Based on LCD: Influence Factors of the Film Printability and Received Qualities"

_pharmaceutics, 2023, doi:10.3390/pharmaceutics15030758_

Round 1

Reviewer 1 Report

The present study suggests that using a 3D printing technique based on liquid crystal display (LCD) can be an effective method for fabricating mucoadhesive oral films for drug delivery. The study investigated the influence of printing formulation and parameters on the formability of the oral films and found that the optimized formulation and parameters resulted in the successful printing of bilayer oral films with desired characteristics. These characteristics include appearance and dimensions, mechanical property, adhesion ability, physical state, drug release, and in vivo therapeutic efficacy. The results indicate that LCD-based 3D printing is a promising alternative to traditional manufacturing approaches and could offer personalized customization for oral mucosal drug delivery.

There may be several limitations to the findings presented in this study:

1.      The sample size used is not large enough, the authors need to explain clear how statical analysis was performed and how the results are significant.

2.      Tartrazine was used as the model drug rational needs to provided for this selection further studies are needed to evaluate the compatibility of different types of drugs with the oral films – this aspect is one of the major limitation.

3.      Manufacturing scalability: The study has used a small-scale manufacturing process, and further studies are needed to determine if the process can be scaled up for large-scale production. The author need to elaborate on this element.

4.      Cost-effectiveness: The cost of using LCD-based 3D printing technology for manufacturing oral films may be higher than traditional manufacturing methods, and further research is needed to determine the cost-effectiveness of this approach.

Minor comments:

1.      Figure 3, 4 – statical analysis need to be performed for A and B.

2.      Figure 13 (B) – scale bar needs to be shown for histological images and magnification details need to be provided. For (A) there is no mention what the error bars are SEM or SD proper statistics need to be shown.

3.      Overall there are some typographical errors these need to be corrected. 

Reviewer 2 Report

The paper deals with the preparation of mucoadhesive films by a photopolymerization-based 3D printing technique, the influence of formulation and printing parameters on the printing formability of the oral films were also studied. The text is well organized and written, but some modifications are required.

Specific comments

·        The abstract is too general. The background is too long, while it lacks of the description of the results and the materials employed for film production. Therefore, the abstract should be reorganized and more detailed information must be included.

·        Line 103, 485, check typing error

·        Can you add a statistical analysis in Table 7 and table 8 ?

·        Why haven’t you studied the effect of drug concentration on printability and on film features? If possible, can you add some data in this regard?

·        What was the name of the commercially available dexamethasone acetate film? Please, include it the text

·        How have you calculated the ulcer surface in a precise manner?

·        In the conclusions the Authors state “ LCD technique is a promising alternative to precisely fabricate oral films”. This sentence should be supported by further investigations i.e.  the uniformity of content assay. This assay should be performed in order to establish the suitability of this technique in a production process.

Round 2

Reviewer 1 Report

I am satisfied with the revision undertaken to address my comments and improve the manuscript 

Reviewer 2 Report

The Authors have satisfied the reviewer's requests, no more comments are needed.